# Long-Term Outcome of Leptospirosis Infection with Acute Kidney Injury

**DOI:** 10.3390/biomedicines10102338

**Published:** 2022-09-20

**Authors:** Chih-Hsiang Chang, Wei-Chiao Sun, Su-Wei Chang, Cheng-Chia Lee, Pei-Chun Fan, Huang-Yu Yang, Chih-Wei Yang

**Affiliations:** 1Kidney Research Center, Department of Nephrology, Chang Gung Memorial Hospital, Linkou Branch, Taoyuan 333, Taiwan; 2Postgraduate Institute of Clinical Medical Science, College of Medicine, Chang Gung University, Taoyuan 333, Taiwan; 3Department of Nephrology, New Taipei Municipal TuCheng Hospital, New Taipei City 236, Taiwan; 4Clinical Informatics and Medical Statistics Research Center, College of Medicine, Chang Gung University, Taoyuan 333, Taiwan; 5Division of Allergy, Asthma, and Rheumatology, Department of Pediatrics, Chang Gung Memorial Hospital, Taoyuan 333, Taiwan

**Keywords:** leptospirosis, acute kidney injury, chronic kidney disease, long-term prognosis

## Abstract

Acute kidney injury (AKI) is associated with long-term mortality and morbidity outcomes. Severe leptospirosis usually results in AKI and multiple organ failure, but is associated with favorable short-term outcomes, if treatment is initiated early. However, information on long-term outcomes after leptospirosis-associated AKI is limited. The effects of leptospirosis on resulting chronic kidney disease (CKD), as well as mortality, were evaluated in this study. We studied 2145 patients with leptospirosis from the National Health Insurance Research Database over an 8-year follow-up period. Patient demographics and characteristics were analyzed for AKI and dialysis. The risk factors for renal outcomes were analyzed using multivariate logistic regression. In total, 443 (20.6%) patients had AKI. Among them, 77 (3.6%) patients received replacement therapy (AKI-RRT group). Long-term mortality was higher in the AKI-RRT group than in the AKI group and non-AKI group, based on a multivariate logistic regression model. Similarly, the incidence rate of CKD was highest in the AKI-RRT group, followed by the AKI and non-AKI groups. Leptospirosis, complicated with AKI, may play a critical role in the long-term outcomes, resulting in CKD. The severity of AKI determines the incidence of CKD. Additional prospective investigations for the early detection of AKI in leptospirosis are warranted.

## 1. Introduction

Leptospirosis, caused by the pathogenic spirochete Leptospira, is known as the most widespread zoonotic disease in the world, particularly in tropical and subtropical regions. Human can be infected through water or soil contaminated by the urine of infected animals [1]. Constitutional symptoms such as fever, headache, and myalgia are common in leptospirosis, followed by conjunctival suffusion, cough, nausea, vomiting, diarrhea, splenomegaly, hepatomegaly, lymphadenopathy, pharyngitis, and aseptic meningitis. In severe leptospirosis, acute kidney injury (AKI), jaundice, pulmonary hemorrhage, acute respiratory distress syndrome, uveitis, optic neuritis, myocarditis, and rhabdomyolysis have been reported. AKI caused by leptospirosis is common and presents as tubulointerstitial nephritis [2]. In vitro and animal studies have shown that leptospirosis may induce renal fibrosis [3,4]. Because Leptospira may colonize the proximal tubules and lead to chronic tubulointerstitial nephritis and fibrosis, asymptomatic exposure to Leptospira can be associated with increased risk of chronic kidney disease (CKD). A long-term follow-up study including 44 patients in Sri Lanka revealed that 9% of patients with leptospirosis-related AKI developed early-stage CKD afterward [5]. AKI has been associated with CKD, cardiovascular events, gastrointestinal bleeding, all-cause mortality, and even diabetes [6,7,8]. However, chronic human infection is possible in sporadic cases [9], such as late onset uveitis with leptospiral DNA identified from aqueous humor [10] and central sleep apnea attributed to chronic neuroleptospirosis [11,12]. In patients with leptospirosis, the reports of long-term outcomes are sparse and limited to small-scale observational cohorts. Therefore, in our investigation, the incidence of mortality, CKD, and dialysis were evaluated in a population-based cohort.

## 2. Materials and Methods

### 2.1. Data Source

This retrospective cohort study used claims data from the Taiwan National Health Insurance Research Database (NHIRD). The NHIRD contains health insurance claims data from a single-payer National Health Insurance (NHI) program, which is operated by the Taiwan government, with a coverage rate of over 99% of the total population. The diagnosis and treatment of leptospirosis were performed according to the instruction of the Centers for Disease Control. The treatment of AKI was based on KDIGO guidelines. Disease diagnoses are based on the International Classification of Diseases, Ninth Revision, Clinical Modification (ICD-9-CM). This study was approved by the Chang Gung Memorial Hospital Ethics Review Committee (approval number: 201601781B1).

### 2.2. Patient Selection and Study Design

This study aimed to verify the relationship between AKI and long-term prognosis in patients admitted for leptospirosis. As illustrated in Figure 1, patients aged above 20 years with an inpatient diagnosis of leptospirosis between 2006 and 2013 were sampled from the entire Taiwanese population. The discharge date for leptospirosis was defined as the index date. Patients who received an organ transplant, had underlying cirrhosis, autoimmune disease, or malignancy were excluded. To clarify the relationship between leptospirosis and CKD, patients with prior renal dysfunction were also excluded. To evaluate the long-term outcomes, patients who died during admission or within 90 days after discharge were excluded. Patients were divided into the non-AKI and AKI groups, according to the diagnosis made during the index admission. Patients with AKI were further divided into those who received replacement therapy (RRT) and those who did not (non-RRT), according to the Taiwan NHI reimbursement codes during the index admission.

### 2.3. Covariates and Study Outcomes

Diseases were identified using ICD-9-CM diagnostic codes. The covariates were age, sex, comorbidities, and medications. Comorbidities were identified, when reported, as more than two outpatient visits or one inpatient stay in the past year. All ICD-9-CM code for comorbidities are listed in Appendix A. Medications were identified by prescriptions filled at least twice or prescriptions refilled at least once for a chronic illness in the past three months. Most diagnostic codes used for these comorbidities have been validated in previous NHIRD-based studies [13]. The follow-up outcomes were all-cause mortality, new-onset CKD, and long-term dialysis. All-cause mortality was defined by withdrawal from the NHI program, and renal outcome was identified by at least two outpatient visits or admissions [14]. New-onset CKD was defined by at least two outpatient visits or one inpatient stay. Long-term dialysis was detected using a subset of the NHIRD, the Registry for Catastrophic Illness Patient Database.

### 2.4. Statistical Analysis

We analyzed patient characteristics among the study groups using one-way analysis of variance for continuous variables and the chi-square test for categorical variables. Pairwise comparisons with Bonferroni adjustments were performed if the overall difference was significant. The risk of mortality among the study groups was compared using the Cox proportional hazards model. The incidence of new-onset CKD or dialysis in the study groups was analyzed using the Fine-Gray subdistribution hazard model, which considered mortality as a competing risk. Survival analyses were conducted with adjustment for age, sex, comorbid conditions, and index year. A two-sided *p* value of <0.05 was considered statistically significant. Adjustment for multiple testing (multiplicity) was not made in this study. All statistical analyses were performed using SAS software, version 9.4 (SAS Institute, Cary, NC, USA).

## 3. Results

### 3.1. Study Population Characteristics

In total, 2145 patients were included in this study. The mean age was 48.99 ± 15.20 years, and 69.79% of the patients were male. AKI with dialysis was confirmed in 77 (3.5%) patients. The baseline demographic and clinical characteristics are listed in Table 1. Patients who had AKI or received dialysis differed substantially in age, sex, prevalence of hypertension, and coronary artery disease from patients without AKI. They also received more angiotensin-converting enzyme inhibitors/angiotensin II receptor blockers (ACEis/ARBs) and more oral antihyperglycemic agents than the patients without AKI. Otherwise, the prevalence of diabetes mellitus, dyslipidemia, and gout, as well as the use of nonsteroidal anti-inflammatory agents, were similar between both groups. The distribution of renal complications by region in Taiwan is presented in Figure 2. No regional difference across Taiwan was observed. According to Figure 3, a peak occurred in 2009 because of Typhoon Morak [15]. Otherwise, the percentage of AKI in patients with leptospirosis was equally distributed across the years.

### 3.2. Patient Outcome among Various Study Groups

The mean follow-up time in this study was 4.3 years (standard deviation 2.2 years). The number and event rate of long-term outcomes in each study group are listed in Table 2. A rising trend was observed in all-cause mortality, the incidence of new-onset CKD, and end-stage renal disease (ESRD) in the non-AKI, AKI, and dialysis groups. The event rates of all-cause mortality and new-onset CKD were highest in patients in the AKI-RRT subgroup. Nevertheless, the incidence of ESRD was zero in the non-AKI and AKI groups.

Table 3 presents the results from the pairwise comparison of follow-up outcomes among the study groups after adjustment for covariates. Model 1 includes the associated baseline characteristics, and the Model 2 further includes the medications associated with adverse renal outcomes. No significant differences were observed in the risk of mortality between the AKI non-RRT subgroup and the non-AKI group (hazard ratio [HR] 1.19; 95% confidence interval [CI] 0.83–1.72). However, the risk of all-cause mortality was significantly higher in the AKI-RRT subgroup than in the other 2 groups. The incidence of CKD was significantly higher in the 2 AKI subgroups than in the non-AKI group. However, no difference was observed between the 2 AKI subgroups. Figure 4 depicts the unadjusted cumulative event rates of mortality and new-onset CKD in each study group.

## 4. Discussion

Most cases of leptospirosis are mild and present as subclinical symptoms; consequently, diagnosis becomes difficult and is easily overlooked. Therefore, the prevalence of leptospirosis infection is presumed to be underestimated. The reported data in this study may come from various diagnostic approaches, as some diagnoses were based on clinical manifestations and were confirmed by a microscopic agglutination test (MAT) or a polymerase chain reaction test. Leptospirosis is mostly reported in tropical and subtropical regions, especially following cyclones and flooding. Moreover, leptospirosis is more common in workers exposed to infected animals and contaminated water, such as paddy workers, animal handlers, mine workers, sewer workers, and sugarcane workers [16]. A global survey performed from 2007 to 2013 revealed that the area with the highest risk is Middle America (63%), followed by the Western Pacific (15%), Southeast Asia (14%), and Europe (8%) [17]. Although the African (1%) and Eastern Mediterranean regions (0.5%) do not present numerous leptospirosis cases, possible underdiagnosis was suspected because of the limited diagnostic capabilities. In Taiwan, 251 patients were seropositive out of 1632 suspected cases from 2000 to 2002. Finally, 57 cases were confirmed to be leptospirosis; the mean age of patients was 45 years, and most were male (89.47%) [18]. Several outbreaks have been reported in Taiwan following cyclones and flooding. Although all patients with a confirmed leptospirosis diagnosis were admitted with multiple organ failure, the disease was identified early and they were treated with adequate antibiotics, and only 1 patient died in 2001.

In vitro and animal studies have shown that *Leptospira* infection can lead to renal fibrosis [3,4]. Because *Leptospira* may colonize the renal proximal tubules and cause chronic tubulointerstitial nephritis and fibrosis, asymptomatic exposure to *Leptospira* may induce a risk of CKD. A long-term follow-up study including 44 patients in Sri Lanka revealed that 9% of patients with leptospirosis-related AKI developed early-stage CKD afterward [5]. In southern Taiwan, where flooding is frequent because of cyclones, 33.2% of 3045 survey participants were positive for the anti-Leptospira antibody. The estimated glomerular filtration rate (eGFR) of the seropositive individuals was lower by 2.5%, but the percentage of patients with stage 3a-5 CKD was higher (14.4% vs 8.5%) than that of the seronegative individuals [19]. In endemic regions of leptospirosis, individuals with higher anti-Leptospira antibody MAT titers tend to have a lower eGFR and higher urinary kidney injury molecule-1 to creatinine ratios. Notably, two individuals with persistently high MAT titers and detectable urine leptospiral DNA showed significantly deteriorating renal function during a 2-year follow-up, which suggested that asymptomatic leptospiral kidney colonization may cause CKD [20]. It has been known that leptospires are shed from the urine after the patients recover from acute illness, and asymptomatic leptospiral infection can result in high disease transmission rate. In Nicaragua, during a major outbreak of leptospirosis in 1995, 15.0% of patients were reported to be seropositive. Only 29.4% of them presented with preceding febrile illness, and the remaining 70.6% had asymptomatic infection [21]. In the Amazonian region, 4.1% of Peruvian residents carry *Leptospira* asymptomatically [22]. In Tamil Nadu, India, positive urine leptospiral DNA was identified in 52.7% of asymptomatic individuals. Among those with positive urine leptospiral DNA, 20.4% were negative for MAT, suggesting that immune deficiency may contribute to the persistence of carrier status [23]. Large mammals with the carrier status, including humans, can excrete considerably more urine and shed considerably higher concentrations of *Leptospira* per day than rodents [24]. Therefore, in regions where leptospirosis is endemic, the possibility that humans are maintenance hosts, rather than incidental hosts, is high. The prevalence of carrier status for chronic leptospirosis and asymptomatic infection is often overlooked.

AKI not only worsens short-term outcomes, but also compromises long-term prognosis [25,26]. A cohort study conducted in Canada included 3769 adults with AKI requiring dialysis to assess the primary outcome of chronic dialysis during a 3-year follow-up. The adjusted HR of chronic dialysis after AKI requiring dialysis was 3.23 (95%CI, 2.70–3.86). The absolute risk for ESRD was increased by 10-fold in patients with normal baseline eGFR, and approximately 2-fold in patients with decreased baseline eGFR, for those with AKI [27]. The risks of CKD and ESRD were proportional to the severity of AKI [28]. Coca et al. included 13 cohort studies to evaluate the long-term renal and nonrenal outcomes in patients with AKI. Patients with AKI had a higher risk of CKD (pooled adjusted HR 8.8; 95% CI 3.1–25.5), ESRD (pooled adjusted HR 3.1; 95% CI 1.9–5.0), and mortality (pooled adjusted HR 2.0; 95% CI 1.3–3.1) than did patients without AKI [6]. AKI was also a risk factor for new-onset or deteriorating proteinuria, suggesting a possible mechanism linking AKI and subsequent CKD [29,30]. However, only one small-scale study demonstrated that 9% of patients with leptospirosis developed early-stage CKD [5]. More recently, the impact of Leptospira on CKD was extended to the understanding of the cause of CKD of unknown etiology. The asymptomatic carrier, or debris from Leptospira retained in infected kidneys, might be one of the mechanisms for chronic inflammation that lead to CKD [31,32,33,34].

In this study, using the NHIRD, outcomes were evaluated based on the confirmed diagnosis of leptospirosis, and the severity of AKI determined the CKD outcome. Therefore, this population-based study with an 8-year follow-up period provided the evidence that patients with AKI after leptospirosis infection are prone to develop CKD and require dialysis. Additionally, the mortality rate in patients with AKI after leptospirosis was higher in patients with CKD than in those without CKD in the follow-up years. In the future, evaluating the subclinical groups of infection in which the clinical manifestations of leptospirosis are not evident, as well as the effect of leptospirosis on the incidence of CKD, is necessary.

Despite the satisfactory results obtained in this study, several major limitations should be considered. First, this was an observational cohort study, and the diagnosis was extracted from the NHI claims data, which do not cover biochemistry data or the results from serology examinations. In the absence of detailed information of the serum creatinine level and urine amount, we can only describe the severity of AKI through the presence of renal replacement therapy, or through staging systems, such as the KDIGO or AKIN methods. Moreover, slight, transient elevations of the serum creatinine level might have been missed, and the diagnosis of AKI may thus be underestimated. In addition, there is no information regarding the time between diagnosis with AKI and referral. Further research using experiment models or human studies is warranted to explore the pathophysiology and potential treatments after Leptospira infection.

## 5. Conclusions

Leptospirosis, complicated with AKI, is associated with increased risk of future CKD and dialysis. Dialysis-requiring AKI, compared to non-dialysis-requiring AKI, has a higher risk of poor kidney outcome. Moreover, in patients with leptospirosis-related AKI, the mortality rate is higher in patients with subsequent CKD than in those without subsequent CKD. Additional studies on the timing of leptospirosis identification are warranted to improve the outcomes.

## Figures and Tables

**Figure 1 biomedicines-10-02338-f001:**
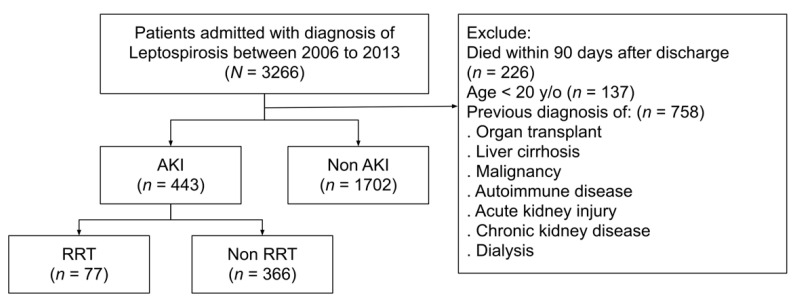
Selection of the study patients.

**Figure 2 biomedicines-10-02338-f002:**
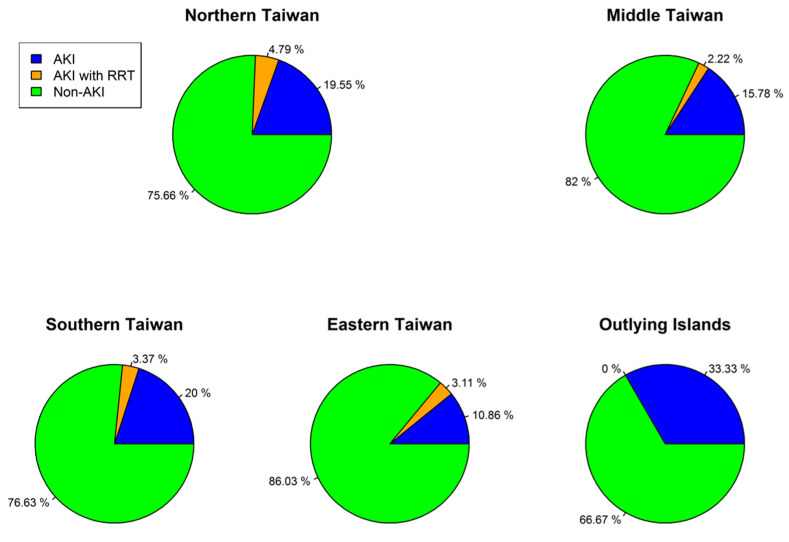
Distribution of Renal Complications by Region in Taiwan.

**Figure 3 biomedicines-10-02338-f003:**
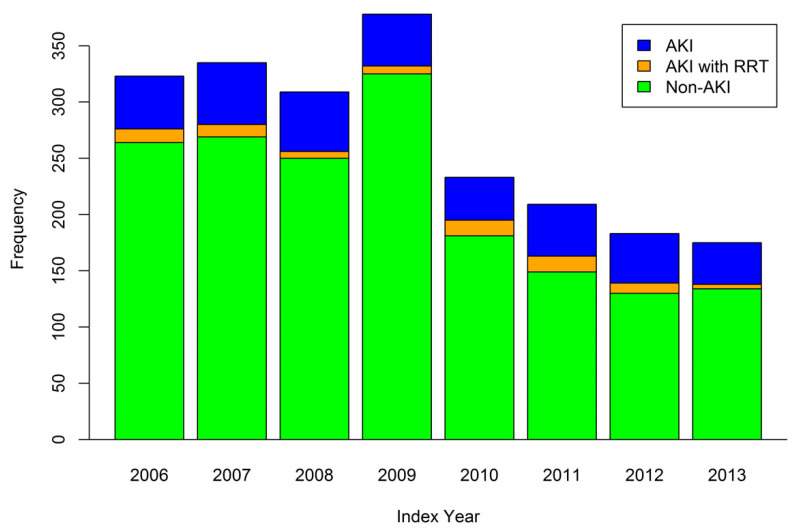
Distribution of the study groups by index year.

**Figure 4 biomedicines-10-02338-f004:**
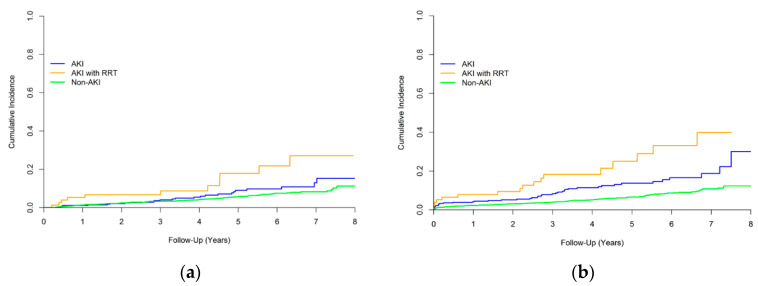
Unadjusted cumulative event rates of (**a**) mortality and (**b**) new-onset CKD in each study group.

**Table 1 biomedicines-10-02338-t001:** Baseline patient profiles.

Variable	Total (n = 2145)	Non-AKI (n = 1702)	AKI (n = 366)	AKI + RRT (n = 77)	*p*
Age (years)	49.0 ± 15.2	48.1 ± 15.1	51.6 ± 15.3	55.4 ± 14.9	<0.001
Male	1497	1145	297	55	<0.001 †
Risk factors for AKI					
Diabetes mellitus	364	281	64	19	0.462
Hypertension	683	518	128	37	0.031 *
Hyperlipidemia	512	403	85	24	0.952
Gout	405	304	82	19	0.028 †
PAOD	85	64	16	5	0.471
CAD	325	253	52	20	0.91 *
Heart failure	114	84	23	7	0.205
Stroke	181	137	34	10	0.319
Medication					
ACEI/ARB	276	200	60	16	0.007 †
Diuretics	113	86	20	7	0.577
OHA	175	134	29	12	0.678 *
Insulin	94	66	20	8	0.081
NSAID	322	264	51	7	0.322

Categorical variables are expressed as a number (percentage) and continuous variables as mean ± SD. * AKI vs. AKI-RRT, † Non-AKI vs. AKI. PAOD, peripheral arterial occlusive disease. CAD, coronary artery disease. ACEI, angiotensin-converting enzyme inhibitor. ARB, angiotensin II receptor blocker. OHA, oral hypoglycemic agent. NSAID, non-steroidal anti-inflammatory drug.

**Table 2 biomedicines-10-02338-t002:** Number and percentage of event of outcomes.

Variable	Total (n = 2145)	Non-AKI (n = 1702)	AKI (n = 366)	AKI + RRT (n = 77)
All-cause mortality	182	119	46	17
CKD	282	119	127	36
ESRD/Dialysis	4	0	0	4

CKD, chronic kidney disease. ESRD, end-stage renal disease.

**Table 3 biomedicines-10-02338-t003:** Pairwise comparison of follow-up outcomes according to the study groups.

	HR (95% CI), *p* Value					
	AKI without RRT vs. Non−AKI		AKI with RRT vs. Non−AKI		AKI with RRT vs. AKI without RRT	
Outcomes/Model	HR (95% CI)	*p*	HR (95% CI)	*p*	HR (95% CI)	*p*
All-cause mortality						
Unadjusted	2.07 (1.48−2.92)	<0.001	3.69 (2.22−6.13)	<0.001	1.86 (1.06−3.25)	0.0300
Model 1	1.25 (0.88−1.79)	0.2139	2.15 (1.26−3.68)	0.0051	2.13 (1.15−3.96)	0.0162
Model 2	1.19 (0.83−1.72)	0.3506	2.10 (1.21−3.65)	0.0086	3.01 (1.53−5.93)	0.0015
Chronic kidney disease §						
Unadjusted	6.85 (5.33−8.81)	<0.001	10.56 (7.24−15.39)	<0.001	1.41 (0.97−2.05)	0.0688
Model 1	6.29 (4.86−8.15)	<0.001	7.79 (5.29−11.47)	<0.001	1.23 (0.83−1.81)	0.2967
Model 2	6.27 (4.83−8.13)	<0.001	8.46 (5.67−12.61)	<0.001	1.27 (0.85−1.90)	0.2498

HR, hazard ratio; CI, confidence interval. Model 1 adjusted for baseline age, gender, comorbid conditions listed in Table 1, and study year. Model 2 further adjusted for baseline medications. § estimated using a subdistribution hazard model, in which death was considered as a competing risk.

## Data Availability

Not applicable.

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
