# Peer review of "Long-Term Outcome of Leptospirosis Infection with Acute Kidney Injury"

_biomedicines, 2022, doi:10.3390/biomedicines10102338_

Round 1

Reviewer 1 Report

-         In the introduction part- the authors havent described thoroughly the pathophisiological mechanism of AKI in leptospirosis  and CKD in silent leptospirosis and in those with AKI. Also, the authors haven’t mentioned which other organs are affected by leptospirosis.

-         In the results part, the authors need to provide more explanation related to the models they have choosen (model 1, model 2).

-          the discussions part need to be improve- the authors need to compare their results with other studies.

-         The conclusions part need also to be improved- is to short and doesnt offer enough information for a conclusions part. The authors need to write why this study is important for practioners and for literature. 

Reviewer 2 Report

Interesting study on infection-related AKI.

Nevertheless, that a lack of association between the  risk of AKI and chronic co-morbidities was shown (table 1), the reader (not familiar with leptospirosis cases)  will appreciate some more data on acute risk factors of AKI (especially severe AKI) identified among patients infected with Leptospira.

Specific questions:

- Was the time from the  infection diagnosis to referral to the hospital and to presenting AKI - longer, than in patients with no AKI?

- What was specific treatment (if any) for infection-related TIN (tubular-intestistial nephritis) ?

- Any steroids?

- Any differences in treatment protocol between AKI/AKI-RRT/non-AKI patients?

- Is the common protocol of management universal in all medical centers, or is locally different?

Round 2

Reviewer 1 Report

I dont have anything to add.

Author Response

Thank you for your suggestion.

Reviewer 2 Report

The revised manuscript responses to the previous comment

Author Response

Thank you for your suggestion.